# The Mayan Tropical Rainforest: An Uncharted Reservoir of Tritrophic Host-Fruit Fly-Parasitoid Interactions

**DOI:** 10.3390/insects11080495

**Published:** 2020-08-03

**Authors:** Maurilio López-Ortega, Francisco Díaz-Fleischer, Jaime C. Piñero, José René Valdez-Lazalde, Manuel Hernández-Ortiz, Vicente Hernández-Ortiz

**Affiliations:** 1Instituto de Biotecnología y Ecología Aplicada (INBIOTECA), Universidad Veracruzana, Xalapa, 91090 Veracruz, Mexico; fradiaz@uv.mx (F.D.-F.); manuelhdezo94@gmail.com (M.H.-O.); 2Stockbridge School of Agriculture, University of Massachusetts, Amherst, MA 01003, USA; jpinero@umass.edu; 3Colegio de Postgraduados, Postgrado en Ciencias Forestales, Montecillo, 56230 Texcoco, Mexico; valdez@colpos.mx; 4Red de Interacciones Multitróficas, Instituto de Ecología A.C. Xalapa, 91073 Veracruz, Mexico; vicente.hernandez@inecol.mx

**Keywords:** host-plants, *Anastrepha*, tropics, conservation, frugivory, biodiversity

## Abstract

**Simple Summary:**

Tropical rainforest can provide various ecological services to adjacent agricultural environments, including maintaining and amplifying the numbers of beneficial insects. However, forest fragmentation and the selective cutting of indigenous trees used by native species of fruit flies and their parasitoids, degrades the potential of forests to provide ecological services to agriculture. Over a two–year period, we surveyed natural areas of the Mayan rainforest in Quintana Roo, Mexico. We found 11 species of native fruit flies belonging to the genus *Anastrepha* associated with 25 species of fruits belonging to ten plant families. We report the first records of 10 host plant species of the genus *Anastrepha*. We also report a new undescribed species of *Anastrepha*. The interaction between fruit flies and their parasitoids with host plants depends on fruit availability, which is crucial for the survival of each of these species. Our findings indicate that the areas of the Mayan rainforest surveyed represent a highly important reservoir for the diversity of native parasitoids spatially and temporally that are practically absent in fruits of cultivated plants. Conserving the landscape of the Mayan rainforest is important not only for species conservation, but also for the maintenance of fruit fly host plants of biological control agents in orchard agroecosystems in southeastern Mexico.

**Abstract:**

Over a two–year period, we surveyed natural areas of the Mayan rainforest in Quintana Roo, Mexico. We found 11 species of *Anastrepha* Schiner (Diptera: Tephritidae) infesting 25 species of fruits belonging to ten plant families. We report the first records of 10 host plant species of the genus *Anastrepha*, which include the first report of a plant family (Putranjivaceae) serving as host of *Anastrepha fraterculus* (Wiedemann) infesting *Drypetes lateriflora* (Sw.) Krug and Urb. (Putranjivaceae). *Pouteria reticulata* (Engl.) Eyma (Sapotaceae) was found, for the first time, to be infested by *Anastrepha serpentina* (Wiedemann) and by a new undescribed species of *Anastrepha*. We also report *Casimiroa microcarpa* Lundell (Rutaceae) as a possible ancestral host for the Mexican fruit fly, *Anastrepha ludens* (Loew), in Central America. The family Sapotaceae was the best-represented host group with three fruit fly species recovered: *A. serpentina*, an economically-important species, found in eight host plants, and *A. hamata* and *A. sp.* (new species). We recorded six species of koinobiont parasitoids: *Doryctobracon areolatus* Szepligeti, *Utetes* (*Bracanastrepha*) *anastrephae* Viereck, *Opius hirtus* Fisher, and *Doryctobracon zeteki* Musebeck, (all Braconidae), and *Aganaspis pelleranoi* (Brethés) and *Odontosema anastrephae* Borgmeier, (both Figitidae). All these parasitoid species represent at least a new report for their host plants. Of the whole parasitoid community, *D. areolatus* was the most important parasitoid species with 52.7% of presence in 12 host plant species, parasitizing six fruit fly species. The interaction between fruit flies and their parasitoids with host plants depends on fruit availability, which is crucial for the survival of each of these species. Conserving the landscape of the Mayan rainforest is important not only for species conservation, but also for the maintenance of fruit fly host plants in orchard agroecosystems in southeastern Mexico.

## 1. Introduction

Herbivorous insects have a powerful influence on plant abundance and distribution, as well as on the composition of plant communities [1,2]. The study of interactions between insects and fruits is one of the main challenges for understanding the reproductive success of many angiosperms because the damage caused by insects can cause the abortion of a wide variety of fruits [3,4]. Fruit-eating insects can influence the production of seeds, due to direct damage, and by indirect damage through biochemical changes that cause premature ripening of the fruit or increased protein levels. For example, wild tobacco, *Solanum mauritianum* Scop (Solanaceae), fruits infested by *Dacus cacuminatus* (Hering) (Diptera: Tephritidae) are reported to contain twice the levels of proteins and essential amino acids when compared to uninfested fruits [5]. Despite the ongoing loss of the original habitat in tropical ecosystems and the resulting detrimental effects on biodiversity [6,7], these ecosystems still harbor substantial numbers of potential host plants in practically every biological form, including herbs, vines, shrubs, and trees.

The great diversity of plants that occurs in tropical rainforests suggests the existence of a high diversity of tephritid fruit flies. The Neotropical genus *Anastrepha* exhibits great richness, estimated in 283 known species, some of which are pests of economic significance [8]. In Mexico and various countries of Central and South America, numerous samples of wild and cultivated tropical fruits have been examined in order to determine their seasonal phenology and infestation levels, produced mainly by *Anastrepha* species. In numerous occasions, such insect-plant biological interactions were recorded for the first time [9,10,11,12,13,14,15,16,17]. There are few systematic studies on trophic interactions between frugivorous tephritids and their host plants in natural environments in the Americas [18,19,20,21], unlike those carried out in agroecosystems where only a few fruit fly species are found associated with cultivated fruits [22,23]. There is an implicit depletion of these natural systems, in terms of both taxonomic richness and host-fruit fly interactions, due to the introduction of non-native fruit trees to the Americas, where they have recently experienced a trophic adaptation and new herbivore/parasitoid interactions.

In Mexico, the most recent reports include 39 described species of *Anastrepha*, which inhabit different regions of the country [24]. Some fly species, such as *Anastrepha tehuacana* Norrbom reproduce on plants that are endemic to xeric environments of central Mexico. Currently, *A. tehuacana* is considered a threatened species [16]. The identification of native parasitoids requires an intensive analysis of native and exotic fruits in order to verify the association between fly and parasitoid species and their host plants. A large guild of native parasitoids associated with *Anastrepha* spp. has been discovered in recent years [13,19,20,25,26]. Most parasitoid species identified, thus far belong to the family Braconidae, which are important in the suppression of natural populations of fruit flies and are of great interest in biological control techniques for fruit flies that cause severe economic losses in commercial orchards, due to the specificity to their hosts [27,28,29]. The parasitoid guild known in Mexico comprises 15 genera from six families represented by native species, as well as three exotic species, for which there are reports of proportions of parasitism and the range of hosts used for those fruit fly communities [19,30,31,32,33,34,35].

Recent studies have emphasized the importance of tropical rainforests in relation to trophic interactions among wild fruits, fruit flies, and their native parasitoids in those communities [18,19,36,37]. Studies aimed at increasing our knowledge of the diversity of *Anastrepha* fruit flies in natural habitats are fundamental for a better understanding of ancestral and more recent (i.e., in agroecosystems) trophic interactions. The main objective of this study was to identify the interactions between fruit fly species and their parasitoids in a fragment of the Mayan rainforest in the state of Quintana Roo, in the southeast region of México. We conducted intensive surveys of available fruits present throughout two annual cycles along two transects of the rainforest in order to identify the natural tri-trophic interactions represented by fruit fly species-plant-parasitoids, as well as to estimate their infestation rates and degree of parasitism.

## 2. Materials and Methods

### 2.1. Study Site

The study was conducted in natural areas of the Mayan rainforest in the municipalities of Felipe Carrillo Puerto and José María Morelos, in the state of Quintana Roo (Mexico). The predominant vegetation in these areas is characteristic of medium semi-evergreen forests sensu [38], where two tree layers can be distinguished: An upper layer, with characteristic elements, such as *Pseudobombax ellipticum* (Kunth) Dugand), *Simarouba glauca* DC, *Cedrela odorata* L., *Swietenia macrophylla* King, among others; and a middle layer, where we usually find *Metopium brownei* (Jacq.) Urb.), *Manilkara zapota* (L.) P. Royen, *Sickingia salvadorensis* (Standl.), *Brosimun alicastrum* Sw, *Malmea depressa* (Baill) R.E. Fries, and *Gymmanthes lucida* Swart [39].

The sampling area was established by using two transects of rural pathways and roads. The first transect (length: 154 km) comprised the ejidos of Dzula (19°35′ N, 88°24′ W, 28 masl) and X Pichil (19°46′ N, 88°31′ W, 47 masl), in the municipality of Carrillo Puerto, and the ejidos of San Antonio Tuk (19°45′ N, 88°41′ W, 11 masl) and Xumuluc (19°34′ N, 88°31′ W, 6 masl), in the municipality of José María Morelos (19°44′ N, 88°42′ W, 54 masl) (Figure 1 Transect 1). The second transect (length: 230.5 km) comprised the ejidos of Dzula, Laguna Kana (19°21′ N, 88°24′ W, 40 masl), Santa María (19°21′ N, 88°24′ W, 26 masl), and X-hazil (19°20′ N, 88°07′W, 27 masl), in the municipality of Carrillo Puerto (Figure 1 Transect 2).

### 2.2. Collection and Processing of Fruit Samples

During a biennial period, from March 2015 to December 2017, we carried out monthly samplings of available (ripe or unripe) fruits sampled from native and introduced plants along both transects. For each transect, there were about 10−12 stops, and for each stop, we spent about 90 min searching for available fruit. The fruits were either, cut directly from the plants (whenever possible using a tree pruner with a saw blade attached to a 4-m long wooden pole (Coronatoolsusa.com) or picked up when fallen, due to ripeness or damage by an insect. Fruits sampled were not decomposed or partially eaten by animals (Figure 2A). Each fruit sample was placed inside 50 × 80 cm organdy cloth bag. We also obtained botanical samples for subsequent identification, as well as in situ photographs with a professional camera (Canon EOS 70D, Canon Inc., Tokyo, Japan). 

Each fruit sample was weighed in the nearest location, and placed in plastic containers with a metal grid, which rested on large plastic containers provided with sterilized river sand at the bottom as pupation substrate. These containers were placed on shelves inside an open-sided roofed shack, provided by local cooperators. This structure protected the fruit from rain and direct sunlight. The sand was regularly inspected, and all pupae recovered were placed in 500-mL labeled plastic cups covered with cloth. After one week, the fruit samples and the recovered pupae were transported to the Bioprospecting Laboratory of the Instituto de Biotecnología y Ecología Aplicada (INBIOTECA) (Xalapa, Veracruz, Mexico) for further processing. From each sample, we separated and weighed 40 fruits individually in order to obtain an accurate estimation of the infestation index. Depending on their size, fruits were placed in 100 mL, 250 mL or 500 mL plastic containers containing moist sand as pupation substrate. The containers were covered with a cloth until adult fruit fly or parasitoid emergence. 

Botanical samples were identified by direct comparison with specimens from the herbarium of the Instituto de Ecología AC (INECOL)—XAL (Xalapa, Veracruz) and the Centro de Investigaciones de Yucatán (CICY) (Mérida, Yucatán). Adult fruit flies were identified by VHO (INECOL), while parasitoids were identified with the use of taxonomic keys [40] and with the help of Andrey Khalaim (Zoological Institute of the Russian Academy of Sciences, St. Petersburg, Russia). Updated information on scientific names of host plants was obtained by consulting the Tropicos database [41]. Reference specimens of identified plants were deposited in the XAL herbarium (INECOL). Reference samples of fruit flies were preserved in 70% alcohol and deposited in INECOL and INBIOTECA, while parasitoid samples were deposited in INBIOTECA.

### 2.3. Data Analyses

Each sampled group of fruits of each species was weighted. For each sample, fruit infestation levels were calculated by dividing the total number of pupae obtained from the fruit sample by its total weight. The indexes of infestation by flies and of parasitism were obtained by dividing the total number of adult flies and/or parasitoids that emerged from the pupae by the total number of pupae obtained from the sample and multiplied by 100. All data from the localities were georeferenced, and coordinates were converted from degrees, minutes, and seconds (DMS) to decimal degrees (DD) using the website gps—coordinates.net. We used the DD to construct a transect map with GIS software (ArcMaps, Versión 10.6.1).

## 3. Results

### 3.1. Fruit Fly-Host Plant Interactions

We examined fruit samples from 76 plant species of 31 botanical families, which summed a total of 143.26 Kg. We documented the presence of 11 species of *Anastrepha* infesting 25 fruit species belonging to 10 families (Table 1). 

Our sampling efforts resulted in the first records of 10 plant species and a plant family (Putranjivaceae) serving as new hosts of fruit flies of the genus *Anastrepha*. These plant species are: *Drypetes lateriflora* (Sw.) Krug and Urb. (Putranjivaceae) and *Blomia prisca* (Standl.) Lundell (Sapindaceae), hosts for *Anastrepha fraterculus* (Wiedemann); *Passiflora yucatanensis* Killip *Passiflora serratifolia* L., and *Passiflora foetida* L. (Passifloraceae) infested by *Anastrepha chiclayae* Greene; *Laetia thamnia* L. (Salicaceae), infested by *Anastrepha zuelaniae* Stone; *Pouteria reticulata* (Engl.) Eyma (Sapotaceae), infested by *Anastrepha serpentina* (Wiedemann) and *Anastrepha*. sp. (new species); *Vitex gaumeri* Greenm. (Verbenaceae), infested by *Anastrepha ampliata* Hernández-Ortiz (Figure 2B), and recently cited for Campeche [42]; and *Casimiroa microcarpa* Lundell (Rutaceae) infested by *Anastrepha ludens* (Loew) (Table 2, Figure 3A–D).

Of all the fruit fly species, *A. serpentina* exhibited the highest number of hosts in the region, exploiting up to eight host species, all in the family Sapotaceae. The Mexican fruit fly, *A. ludens*, was found in three species of plants of the family Rutaceae and the West Indian fruit fly, *Anastrepha obliqua* (Macquart), was found in three species of the family Anacardiaceae. The guava fruit fly, *Anastrepha striata* Schiner and *Anastrepha curvicauda* (Gerstaecker) were found in single host plant species: *Psidium guajava* L. (Myrtaceae) and *Carica papaya* L. (Caricaceae), respectively (Table 2).

With respect to host plant phenology, the highest availability of fruits was generally observed in the period of April-July, with the highest number and abundance of fruits recorded during May (18 species). In particular, we observed that the fruits of *M. zapota* were present during the whole annual cycle, whereas the two species of the family Salicaceae, which were hosts for *A. zuelaniae*, showed the shortest fructification periods (Table 2). 

### 3.2. Fruit Infestation and Parasitism Rates

Fruit infestation rates were highly variable between the different hosts, ranging between 0.21 and 19.17 pupae/kg of sampled fruit. The highest infestation levels occurred in *Spondias mombin* L. *B. prisca*, *P. reticulata*, and *Sideroxylon capiri* subsp *tempisque* (Pittier) T.D. Penn. (range: 14.6–19.1 pupae/kg fruit). Of these, *P. reticulata* and *S. capiri* showed the highest infestation index values in relation to the size of the sample, compared to other Sapotaceae species of the genus *Pouteria* who had only 1.2–2.0 pupae/kg fruit. The highest fly emergence value was observed in *P. campechana* Baehni, with 97.44% of the flies emerging from pupae, and the lowest rates were observed in *S. mombin* (47.86%) and *P. reticulata* (47.83%) (see Table 3).

We recorded an overall parasitism rate of 19.51%, which means that, at the community level, the fly/parasitoid ratio was 5:1. We recorded six species of koinobiont parasitoids: *Doryctobracon areolatus* (Szépligeti), *Doryctobracon zeteki* Musebeck, *Utetes anastrephae* (Viereck), *Opius hirtus* (Fisher) (all Braconidae), *Aganaspis pelleranoi* (Brethés), and *Odontosema anastrephae* Borgmeier (both Figitidae). All these species parasitized larvae that feed on the pulp of the fruit, with the exception of *D. areolatus* and *D. zeteki*, which also parasitized larvae that infest seeds. *Doryctobracon areolatus* was the most important parasitoid species in the whole community, representing 52.7% of all the recorded parasitoids in terms of abundance. It was present in 12 different species of host plants, parasitizing six species of fruit flies.

The percentage of parasitism fluctuated between 3.49% (for *A. hamata* (Loew) feeding on *P. campechiana*) and 35.54%, (for *A. serpentina* feeding on *P. reticulata*). In the case of the species *A. ludens* and *A. chiclayae*, we did not observe any parasitism. Other species, such as *U. anastrephae*, *A. pelleranoi*, and *O. hirtus*, showed moderate activity, with parasitism rates ranging between 13.8 and 17.8%. The least frequently recovered parasitoid species were *O. anastrephae* and *D. zeteki*, with 0.6 and 0.7% of parasitism, respectively (Table 4). The fly species with the highest richness of parasitoids were *A. fraterculus* and *A. serpentina*, with five species each; in contrast, *A. hamata* and *A.* sp. had only one species each, independently of the number of hosts occupied.

## 4. Discussion

Previous studies reported 39 known described species of *Anastrepha* in Mexico [24,43], including recent records of *Anastrepha tehuacana* Norrbom [16] and *Anastrepha furcata* Lima [44]. For the state of Quintana Roo, there are currently 12 known species of *Anastrepha* [24,36,43,45,46]. The present study contributes with the first records of four additional species: *Anastrepha chiclayae* Greene, *Anastrepha zuelaniae* Stone, *Anastrepha curvicauda*, and a newly discovered species, *Anastrepha* sp., increasing the number of known *Anastrepha* species for the state of Quintana Roo to 16. 

In addition, here we document the first records of eight new hosts plants for fruit flies, including the family Putranjivaceae for the first time. Furthermore, 15 hosts are reported for the first time in Quintana Roo. These results highlight the importance of increasing our knowledge about fruit fly/host plant interactions in natural environments. The Mayan rainforest in the southeast of Mexico constitutes a reservoir for tropical biodiversity, including interactions between fruit flies and their natural enemies. Even though the traditional use of protein-baited traps is important to provide data on the presence and abundance of *Anastrepha* species in a particular region, species richness is higher for native fruits [47].

A noteworthy result is the exploitation of alternative hosts by species of economic significance such as *A. ludens*. In Mexico, this fly species has been reported feeding on approximately 23 host plants, most of them being exotic cultivated species, such as *Citrus* spp. [24]. Two important native host plant species are *Casimiroa greggii* (S. Watson) F. Chiang and *Casimiroa edulis* Llave et Lex [48,49] (both Rutaceae). In the present study, *A. ludens* was recovered from fruits of two *Citrus* species and from *Casimiroa microcarpa* Lundell. The latter plant species is a new record for *A. ludens* in Quintana Roo, which was thought to be restricted to Chiapas and Guatemala [50]. In *C. microcarpa*, the larvae were found to feed exclusively on the seeds (Figure 3), as previously observed in *C. greggii* [48,51]. These habits suggest that the use of these native wild hosts could have broadened the distribution area of these flies through the colonization of citrus species cultivated in other regions of Mexico, while at the same time retaining their native hosts of the genus *Casimiroa* because of their distribution in the region. For example, *C. greggii* is found in the northeast of Mexico, *C. edulis* is distributed from Mexico to Costa Rica, and *C. microcarpa* is distributed mainly in Guatemala [41].

The center of origin of the family Sapotaceae is tropical America, and plant species belonging to this family are of great importance in the structure of ecosystems and biological diversity with approximately 200 genera and close to 450 species of trees and shrubs [52,53]. In addition, the consumption of their fruits represents a highly profitable market. For example, *Manilkara zapota* L. is native to Yucatán (Mexico) and Guatemala [54], and its fruits, which have high commercial value, occur practically all year round are commonly heavily infested by *A. serpentina*. So these are also significant reservoirs of native parasitoids.

Fruit flies can persist in different types of environments. Generalist species can thrive in a matrix of human use with commercial and backyard fruit orchards, while a part of the population remains and survives within the natural forest. That would be the case of *A. ludens* in *C. microcarpa*, a plant species that maintains viable populations of this fruit fly within their natural habitat. Because 70 percent of herbivore species exhibit a high level of specialization, [55], then knowledge of wild plant species that serve as hosts for specialist fruit flies is relevant. For example, *A. zuelaniae*, *A. ampliata*, *A. chiclayae*, *A. hamata*, and *A*. sp., have a restricted range of plants (families Salicaceae, Verbenaceae, Passifloraceae, and Sapotaceae, respectively) on which they feed. An interesting observation was that the fruits of *Pouteria glomerata* (Miq.) Radlk (Sapotaceae) were only infested by *A. serpentina* in the study area, even though fruits of this plant species have been found to be infested by *Anastrepha aphelocentema* Stone [37]. The absence of the latter species may implicate biogeographic and ecological factors that could be responsible for the presence/absence of certain species in a particular site [56,57].

The high percentages of parasitism observed in this study in hosts, such as *Pouteria reticulata* (Engl.) Eyma, *Spondias mombin* L., and *Vitex gaumeri* Greenm (29.5–35.6%), differ from previous reports for Yucatán that show that parasitism levels are low. For example, Hernández-Ortiz et al. [26] documented an overall parasitism rate of 3.69% from cultivated plants. Such contrasting results suggest that the Mayan rainforest actually constitutes a highly important reservoir for the diversity of native parasitoid species. This study confirms that *Doryctobracon areolatus* (Szépligeti) is the native parasitoid with the highest abundance. This parasitoid species is widely found in Mexico and other countries [31,33,34]. Moreover, we report *Opius hirtus* (Fisher) in five new fruit fly-parasitoid associations, all occurring in native tree species infested by different fly species. This finding highlights the preference of this parasitoid for monophagous fly species attacking comparatively small-sized fruits [19,31].

The presence of the parasitoid *Doryctobracon zeteki* Muesebeck in larvae of *A. hamata* infesting *P. campechiana* shows the occurrence of a parasitoid attacking larvae of a fly species that feeds on seeds. This is the first report of parasitism in *A. hamata*. In the case of *D. zeteki*, this parasitoid species was first recorded in Mexico in association with larvae of *Anastrepha cordata* Aldrich feeding on seeds [19]. However, in countries like Colombia, Costa Rica, Panama, and Venezuela, *D. zeteki* has been reported in *P. guajava* [34], in Sapotaceae species, and in other species where it has been recorded as the most abundant species [25,58]. Interestingly, some fruit fly species, such as *A. serpentina*, infesting *Pouteria sapota* (Jacq.) H.E. Moore and Stearn and *P. glomerata*, and *A. chiclayae*, infesting passion flowers, did not exhibit parasitism in their natural hosts, which could be a result of the large size of the fruits they infest. This could be a defense mechanism, since it would be more difficult for parasitoids to find a host inside large fruits, which has been hypothesized for other frugivorous species of *Anastrepha* [31,59,60].

The exotic parasitoid *Diachasmimorpha longicaudata* (Ashmead) has been successfully established in certain tropical agroecosystems with significant percentages of parasitism [34,61,62,63]. However, the results of the present study showed that this species is not established in this natural region of the Mayan rainforest, and therefore, this introduced species appears to have limitations for its establishment in natural environments [19,26,31]. For example, we did not find it in fruits of *Citrus* spp. infested by *A. ludens*, where it is common in other regions, along with the native species *Doryctobracon crawfordi* (Viereck) [33].

## 5. Conclusions

Our findings shed new light into new host plant association for species of the genus *Anastrepha* and their parasitoids in natural environments, and highlight the importance of tropical rainforests for the conservation of biodiversity. The areas of the Mayan rainforest that still preserve a great part of its original composition and structure exhibit a higher richness of wild fruits, such as those examined in this study. Consequently, this represent a highly important reservoir for the diversity of native parasitoids spatially and temporally that are practically absent in fruits of cultivated plants.

## Figures and Tables

**Figure 1 insects-11-00495-f001:**
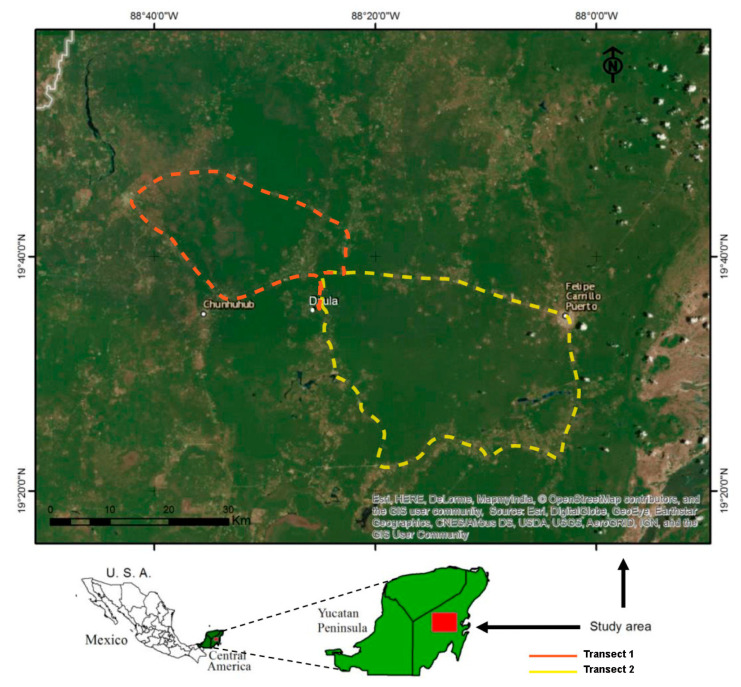
Map showing the location of the study area and an image showing the collection transects (Transect 1: red dashed line and Transect 2: yellow dashed line) in natural areas of the Mayan rainforest in Quintana Roo, Mexico.

**Figure 2 insects-11-00495-f002:**
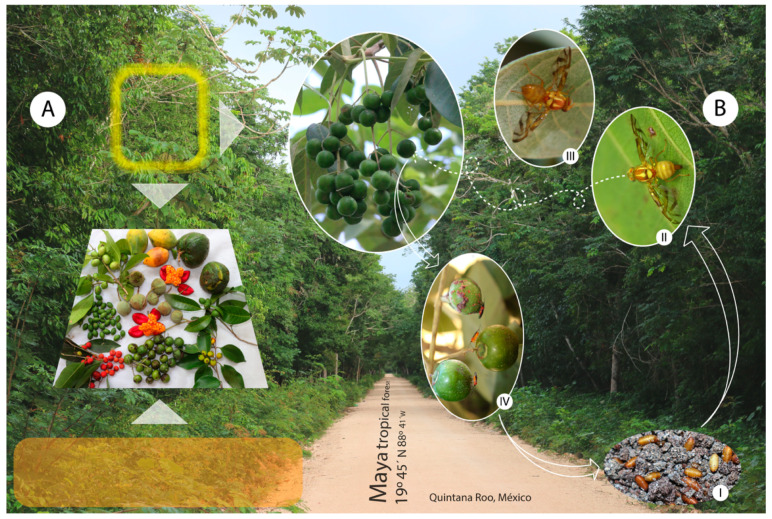
(**A**) Fruit collection methods: Arrows show where fruit was collected above and below the trees. (**B**) Insect life cycle in *Vitex gaumeri* fruits; fruits were found to be infested by *Anastrepha ampliata*. (**BI**) Dipteran larvae emerge from the fruits and fall to the ground in order to bury into the soil to pupate. (**BII**) *A. ampliata* female and (**BIII**) male, (**BIV**) Parasitoid species.

**Figure 3 insects-11-00495-f003:**
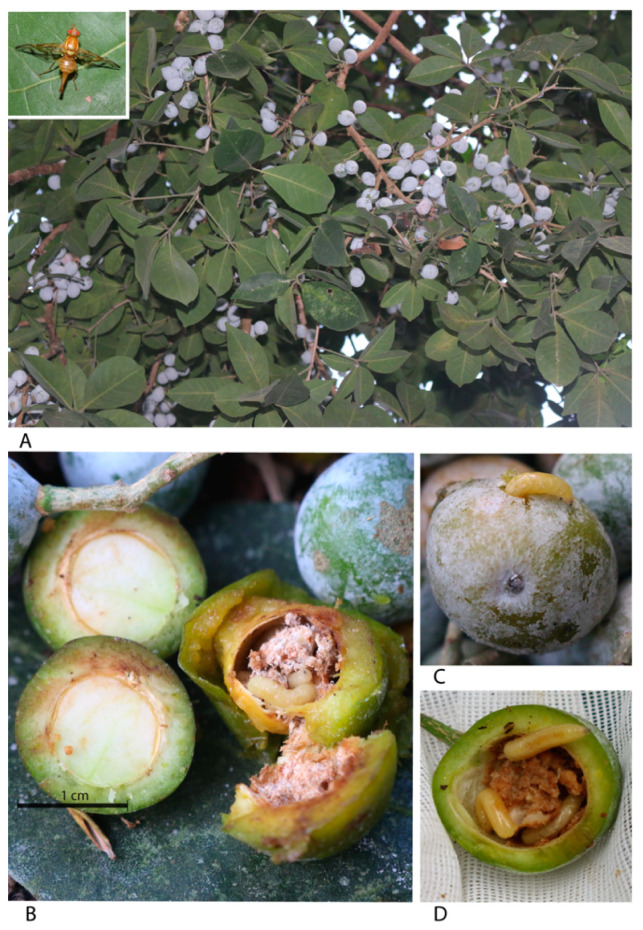
(**A**) Host plant *Casimiroa microcarpa* (first report) with fruits, (**B**) fruits with seeds with and without infestation, (**C**) Larvae of *A. ludens* emerging from the fruits, (**D**) Larvae of *A. ludens* feeding on the seed of *C. microcarpa*.

**Table 1 insects-11-00495-t001:** Host plant family, scientific and local names, and weight of sampled fruit found to be either, infested or uninfested by *Anastrepha* fruit flies during two annual cycles in natural areas of the Mayan rainforest of Quintana Roo, Mexico (March 2015—December 2017).

Plant Family	Scientific Name	Mayan Local Name	Sample Weight (Kg)	Infested Fruit Yes/No
Anacardiaceae	*Metopium brownei* (Jacq.)	Chechen	0.265	N
	*Spondias purpurea* L.	Ciruela	1.100	Y
	*Spondias mombin* L.	Jobo	1.62	Y
	*Mangifera indica* L.	Mango	4.58	Y
Annonaceae	*Annona globiflora* Schlecht.	Anona	0.270	N
	*Annona scleroderma* Saff.	Chujun op	0.680	N
	*Mosannona depressa* (Baill.) Chatrou	Elemuy	0.950	N
Araliaceae	*Dendropanax arboreous* (L.) Decne & Planch.	Sakchaca	0.1620	N
Bixaceae	*Cochlospermum vitifolium* (Willd.) Spreng.	Chuun	0.870	N
Boraginaceae	*Ehretia tinifolia* L.	Beek	0.282	N
	*Cordia dodecandra* DC.	Ciricote	5.310	N
Burseraceae	*Bursera simaruba* (L.) Sarg.	Chaca Rojo	0.176	N
Cannabaceae	*Celtis iguanaea* (Jacq.) Sarg.	Muk	0.475	N
Capparaceae	*Crataeva tapia* L.	Kookche	3.320	N
Caricaceae	*Carica papaya* L.	Chichput	1.500	Y
Ebeneaceae	*Diospyros anisandra* S.F. Blake	Kabche	0.300	N
Euphorbiaceae	*Croton arboreus* Millsp.	Perescuch	0.310	N
	*Gymnanthes lucida* Sw.	Yaiti	1.200	N
Fabaceae	*Caesalpinia platyloba* S. Watson	Chacteviga	0.235	N
	*Platymiscium yucatanum* Standl.	Granadillo	0.340	N
	*Piscidia piscipula* (L.) Sarg.	Jabin	0.410	N
	*Swartzia cubensis* (Britton & Wills) Standl.	Katalox	0.790	N
	*Caesalpinia gaumeri* (Britton & Rose) Greenm.	Kitamche	0.550	N
	*Lysiloma latisiliquum* (L.) Benth.	Tzalam	0.300	N
	*Lonchocarpus yucatanensis* Pittier	Xuul	0.260	N
Lauraceae	*Nectandra salicifolia* (H.B.K.) Nees.	Sakelemuy	1.650	N
Malpighiaceae	*Bunchosia swartziana* Griseb.	Sipche	0.615	N
Malvaceae	*Hampea trilobata* Standl.	Jool	0.520	N
	*Luehea candida* (DC.) Mart.	Kaskaat	0.960	N
	*Pseudobombax ellipticum* (Kunth) Dugand	Amapola	0.700	N
	*Ceiba petandra* (L.) Gaerth.	Yaaxche	1.300	N
Menispermaceae	*Hyperbaena winzerlingii* Standl.	Kekenche	0.173	N
Mimosaceae	*Acacia milleriana* Standl.	Chimay	0.150	N
Moraceae	*Ficus pertusa* L.f.	Juunkiix	1.100	N
	*Brosimum alicastrum* Sw.	Ramon	4.820	N
Myrtaceae	*Psidium sartorianum* (O. Berg) Nied.	Guayabillo	0.885	Y
	*Myrcianthes fragrans* (Sw.) Mc Vaugh	Kojkann	0.312	N
	*Eugenia biflora* (L.) DC.	Pichiche	0.500	N
	*Psidium guajava* L.	Guayaba	1.225	Y
Opiliaceae	*Agonandra macrocarpa* L. O. Williams	Napche	1.765	N
Passifloraceae	*Passiflora foetida* L.	Poochil	0.150	Y
	*Passiflora serratifolia* L.	Maracuya del monte	0.560	Y
	*Passiflora yucatanensis* Killip	Yaax pooch	2.400	Y
Polygonaceae	*Coccoloba acapulcensis* Standl.	Boob/Toyub	0.220	N
Putranjivaceae	*Drypetes lateriflora* (Sw.) Krug & Urb.	Ejuleb	1.910	Y
Rhamnaceae	*Krugiodendrom ferraum* (Vahl) Urb.	Chintoc	0.100	N
Rubiaceae	*Cosmocalyx spectabilis* Standl.	Chactecook	0.164	N
	*Randia truncata* Greenm. & C.H.Thomps.	Kaakalche	0.400	N
	*Exostema mexicanum* A Gray	Sabasche	0.395	N
	*Guettarda combsii* Urb.	Tastab	0.270	N
	*Morinda citrifolia* L.	Noni	3.500	N
Rutaceae	*Citrus aurantium* L.	Naranja agria	6.270	Y
	*Esenbeckia pentaphylla* (Macfad.) Griseb.	Narnaha che	2.630	N
	*Citrus sinensis* (L.) Osbek	Naranja dulce	3.310	Y
	*Casimiroa microcarpa* Lundell	Yuuy	7.300	Y
Salicaceae	*Laetia thamnia* L.	Chauche	3.141	Y
	*Casearia corymbosa* Kunth	Ixiimche	0.424	N
	*Zuelania guidonia* (Sw.) Britton & Millsp.	Tamay	5.672	Y
Sapindaceae	*Blomia prisca* (Standl.) Lundell	Tzol	4.900	Y
	*Cupania belizensis* Standl.	Sal poom	1.200	N
	*Thouinia paucidentata* Radlk.	Kanchunup	0.136	N
	*Melicoccus bijugatus* Jacq.	Guaya	1.200	N
	*Matayba oppositifolia* (A. Rich.) Britton	Ikche	0.370	N
	*Allophylus camptostachys* Radlk.	Kanchunup	0.783	N
	*Talisia oliviformis* (Kunth) Radlk.	Wayum	1.380	N
Sapotaceae	*Manilkara zapota* (L.) Van Royen	Chicozapote	4.200	Y
	*Chrysophyllum cainito* L.	Cayumito	2.800	Y
	*Chrysophyllum mexicanum* Brandegee ex Standl.	Chique	0.690	Y
	*Pouteria campechiana* (Kunth) Baehni	Kaniste	9.393	Y
	*Pouteria sapota* (Jacq.) H. E. Moore and Stearn	Hazz	3.500	Y
	*Sideroxylon capiri* subsp. *tempisque* (Pittier) T.D. Penn.	Subul	6.554	Y
	*Sideroxylon foetidissimum* subsp. *gaumeri* (Pittier) T.D. Penn.	Tsiimimche	0.800	N
	*Pouteria glomerata* (Miq.) Radlk.	Zapote del pueblo	9.935	Y
	*Pouteria reticulata* (Engl.) Eyma	Zapotillo	5.220	Y
Simaroubaceae	*Simarouba glauca* DC.	Paasac	0.360	N
Verbenaceae	*Vitex gaumeri* Greenm.	Yaxnic	9.096	Y

**Table 2 insects-11-00495-t002:** Distribution of the fructification period of plant species sampled from natural areas of the Mayan rainforest of Quintana Roo, Mexico (March 2015—December 2017). Darker shading indicates greater availability of fruits; lighter shading denotes a decreased fruit availability, generally occurring before and after the rainy season. Asterisks indicate new host plant records for *Anastrepha* spp.

Host Family	Host Scientific Name	Fruit Fly Species	Jan	Feb	Mar	Apr	May	Jun	Jul	Aug	Sep	Oct	Nov	Dec
Anacardiaceae	*Mangifera indica* L.	*A. obliqua*					·	·						
	*Spondias purpurea* L.	*A. obliqua*					·	·						
	*Spondias mombin* L.	*A. obliqua*									·	·	·	
Caricaceae	*Carica papaya* L.	*A. curvicauda*			·	·	·	·	·					
Myrtaceae	*Psidium guajava* L.	*A. striata*						·	·	·	·	·		
	*Psidium guajava*	*A. fraterculus*			·	·	·	·	·					
	*Psidium sartorianum* (O. Berg) Nied.	*A. fraterculus*			·	·	·	·	·					
Putranjivaceae	**Drypetes lateriflora* (Sw.) Krug & Urb.	*A. fraterculus*		·	·	·	·	·	·					
Sapindaceae	**Blomia prisca* (Standl.) Lundell	*A. fraterculus*				·	·							
Passifloraceae	**Passiflora yucatanensis* Killip	*A. chiclayae*		·	·	·	·							·
	**Passiflora serratifolia* L.	*A. chiclayae*			·	·	·	·						
	**Passiflora foetida* L.	*A. chiclayae*							·	·	·			
Rutaceae	*Citrus aurantium* L.	*A. ludens*	·	·	·							·	·	·
	*Citrus sinensis* (L.) Osbek	*A. ludens*	·	·									·	·
	**Casimiroa microcarpa* Lundell	*A. ludens*			·	·	·							
Salicaceae	**Laetia thamnia* L.	*A. zuelaniae*							·	·	·			
	*Zuelania guidonia* (Sw.) Britton & Millsp.	*A. zuelaniae*						·	·	·				
Sapotaceae	*Chrysophyllum mexicanum* Brandegee ex Standl.	*A. serpentina*	·	·	·									
	*Chrysophyllum cainito* L.	*A. serpentina*			·	·	·							
	*Manilkara zapota* (L.) Van Royen	*A. serpentina*	·	·	·	·	·	·	·	·	·	·	·	·
	*Pouteria campechiana* Baehni	*A. serpentina*			·	·	·		·	·	·	·	·	·
	*Pouteria glomerata* (Miq.) Radlk.	*A. serpentina*					·	·	·	·	·	·	·	
	**Pouteria reticulata* (Engl.) Eyma	*A. serpentina*				·	·	·	·			·	·	
	*Pouteria sapota* (Jacq.) H.E. Moore and Stearn.	*A. serpentina*								·	·			
	*Sideroxylon capiri* subsp. *tempisque* (Pittier) T.D. Penn.	*A. serpentina*			·	·	·	·						
	*Pouteria campechiana* Baehni	*A. hamata*			·	·	·		·	·	·	·	·	·
	**Pouteria reticulata*	*Anastrepha* sp. 1				·	·	·	·			·	·	
Verbenaceae	**Vitex gaumeri* Greenm.	*A. ampliata*						·	·	·	·	·	·	

**Table 3 insects-11-00495-t003:** Fruit fly species of the genus *Anastrepha*, and their infestation levels and biological data, found in plant species associated as their native and introduced hosts in the sampled region.

Host Family	Host Scientific Name	Fruit Fly Species	Recovered Pupae	Number of Pupae/Kg of Fruit	Sex Ratio (F/M)	Emergence %
Anacardiaceae	*Mangifera indica*	*A. obliqua* (Macquart)	55	1.20	28/19	85.45
	*Spondias mombin*	*A. obliqua*	280	17.28	73/61	47.86
	*Spondias purpurea*	*A. obliqua*	106	9.64	37/49	81.13
Caricaceae	*Carica papaya*	*A. curvicauda* (Gerstaecker)	83	5.53	31/29	72.29
Myrtaceae	*Psidium guajava*	*A. striata* Schiner	92	7.51	4/5	9.78
		*A. fraterculus* (Wiedemann)		--	19/16	38.04
	*Psidium sartorianum*	*A. fraterculus*	21	2.37	14/3	80.95
Putranjivaceae	*Drypetes lateriflora*	*A. fraterculus*	142	7.43	43/47	63.38
Sapindaceae	*Blomia prisca*	*A. fraterculus*	720	14.69	358/302	91.67
Passifloraceae	*Passiflora foetida*	*A. chiclayae* Greene	9	6.00	4/3	77.78
	*Passiflora serratifolia*	*A. chiclayae*	37	6.61	21/15	97.3
	*Passiflora yucatanensis*	*A. chiclayae*	5	0.21	3/1	80.0
Rutaceae	*Citrus aurantium*	*A. ludens* (Loew)	278	4.43	128/123	90.29
	*Citrus sinensis*	*A. ludens*	68	2.05	31/25	82.35
	*Casimiroa microcarpa*	*A. ludens*	383	5.24	191/162	92.17
Salicaceae	*Laetia thamnia*	*A. zuelaniae* Stone	199	6.34	89/76	82.91
	*Zuelania guidonia*	*A. zuelaniae*	180	3.17	61/52	62.78
Sapotaceae	*Chrysophyllum cainito*	*A. serpentina* (Wiedemann)	64	2.28	36/19	85.94
	*Chrysophyllum mexicanum*	*A. serpentina*	12	1.74	5/6	91.67
	*Manilkara zapota*	*A. serpentina*	342	8.14	110/129	71.13
	*Pouteria campechiana*	*A. serpentina*	117	1.25	64/50	97.44
	*Pouteria glomerata*	*A. serpentina*	126	1.27	60/57	92.86
	*Pouteria sapota*	*A. serpentina*	70	2.00	24/35	84.29
	*Pouteria reticulata*	*A. serpentina*	738	19.17	185/168	47.83
	*Sideroxylon capiri* subsp. *tempisque*	*A. serpentina*	1195	18.23	605/433	86.86
	*Pouteria campechiana*	*A. hamata* (Loew)	172	1.83	95/68	94.77
	*Pouteria reticulata*	*Anastrepha* sp. 1	265	6.88	102/116	82.26
Verbenaceae	*Vitex gaumeri*	*A. ampliata* Hernández-Ortiz	633	6.96	197/216	65.24

**Table 4 insects-11-00495-t004:** Parasitoid species and levels of parasitism of fruit fly species of the genus *Anastrepha* in their native and introduced hosts in the sampled region.

Family	Host Plant Scientific Name	*Anastrepha* Species	Recovered Fruit Fly Pupae	Parasitoid Species	Parasitoid Sex Ratio (F/M)	Total No. Parasitoids	% Parasitism
Anacardiaceae	*Mangifera indica*	*A. obliqua*	55	*Doryctobracon areolatus* (Szépligeti)	1/2	3	5.45
	*Spondias mombin*		280	*D. areolatus*	38/25		
				*Utetes anastrephae* (Viereck)	15/21	99	35.36
	*Spondias purpurea*		106	*D. areolatus*	10/8	18	16.98
Myrtaceae	*Psidium guajava*	*A. fraterculus*	92	*D. areolatus*	8/5		
				*Aganaspis pelleranoi* (Brethes)	7/2		
				*Odontosema anastrephae* Borgmeier	2/1	25	27.17
Putranjivaceae	*Drypetes lateriflora*		142	*D. areolatus*	5/3		
				*U. anastrephae*	3/4		
				*Opius hirtus* (Fisher)	2/1		
				*A. pelleranoi*	8/12	38	26.76
Sapindaceae	*Blomia prisca*		720	*D. areolatus*	27/30		
				*U. anastrephae*	12/11		
				*O. hirtus*	7/4	91	12.64
Salicaceae	*Laetia thamnia*	*A. zuelaniae*	199	*D. areolatus*	13/17		
				*O. hirtus*	2/0	32	16.08
	*Zuelania guidonia*		180	*D. areolatus*	12/7		
				*A. pelleranoi*	4/3	26	14.44
Sapotaceae	*Manilkara zapota*	*A. serpentina*	342	*D. areolatus*	16/17		
				*O. hirtus*	4/2		
				*A. pelleranoi*	22/15		
				*O. anastrephae*	3/1	80	23.81
	*Pouteria reticulata*		738	*D. areolatus*	29/31		
				*U. anastrephae*	48/38		
				*O. hirtus*	28/21		
				*A. pelleranoi*	37/21	263	35.54
	*Sideroxylon capiri* subsp. *tempisque*		1195	*D. areolatus*	35/39		
				*O. hirtus*	11/4		
				*A. pelleranoi*	1/2	112	9.37
Sapotaceae	*Pouteria campechiana*	*A. hamata*	172	*Doryctobracon zeteki* Musebeck	5/1	6	3.49
	*Pouteria reticulata*	*Anastrepha* sp.	265	*D. areolatus*	7/12		7.17
Verbenaceae	*Vitex gaumeri*	*A. ampliata*	633	*D. areolatus*	58/51		
				*U. anastrephae*	15/11		
				*O. hirtus*	27/25	187	29.54

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
