# Peer review of "The Mayan Tropical Rainforest: An Uncharted Reservoir of Tritrophic Host-Fruit Fly-Parasitoid Interactions"

_insects, 2020, doi:10.3390/insects11080495_

Round 1
Reviewer 1 Report
This paper is a two year study of species richness and phenology of Anastrepha fruitfly and their parasitoids in Mayan tropical rainforest. While the result seems to have novelty with new species and new trophic relations of species, the sampling strategy could be described more precisely.
Specific comments
Section 2.2., how did you determine the sampling amount of fruits for different species with different amounts of fruits over time? For example, some proportion of mature fruits or fixed number of fruits?
The order and family of Anastrepha should be mentioned at first appearance.
Author Response
Please see the attachment,
Thank you

Reviewer 2 Report
The paper entitled „The Mayan tropical rainforest: an uncharted reservoir of tritrophic host-fruit fly-parasitoid interactions” reports on the results of a two-year survey of tephritid flies, their host plants and parasitoids. It contributes to a body of knowledge about biodiversity in a tropical rainforest and sheds light on natural tritrophic interactions of a group of fruit flies that contains agricultural pest species and therefore even has potential further practical applications.
The manuscript is well-written, but could use some more detail in the material and methods section as well as a slight improvement in the presentation of results (see comments below).
Line by line comments:
17: add taxonomic information about “Anastrepha”
19: see comment above
20: add taxonomic information about “Drypetes laterifolia” and “Pouteria reticulata”
22: see comments above, add for whole abstract
35: remove “trophic interactions”, “parasitoids” and “Mayan rainforest” as keywords, exchange for some not mentioned in the title
39: change “herbivore” to “herbivorous”
45: add taxonomic information for wild tobacco
56-59: After this sentence you could go into a little more detail about the differences between natural and agroecosystems in terms of biodiversity and specifically the group of fruit flies and their parasitoids and why it is important to study natural ecosystems to justify your study.
61: add comma before “such as”
62: remove “s” in “reproduces”
62: mention species name again instead of “it” – or change the sentence structure
69: Remove “in this sense”
82: You might want to think about moving the information in brackets into the material and methods section.
94: Is there a justification for those two transects? Did you expect differences between them?
Figure 1 legend: What do the “T1” and “T2” refer to? Is it “transect”? In the figure they are “route 1” and “route 2”? Best be consistent.
107: remove comma
108-109: Did you record where on the transect you took the samples? Or did you just combine everything from one transect? I see you refer to this later in line 140, but I could not find any mention in the results. Did you analyze the geospatial data?
109: change to “a 50x80 cm organdy cloth bag.”
119: any information about the conditions in the house? Temperature, humidity?
120: How long were the samples stored in the house before being transported to the institute?
122: What constitutes as one sample? Above you mention single fruits, here you seem refer to larger quantities. Please give a definition.
124: replace “and” with “or”
136: I don’t understand that sentence.
136-138: This is by the total number of fruits you sampled of one plant species? Or per actual fruit and then later averaged?
149: remove “new”
150: replace “such” with “these”
Table 1: The “sample weight” is the total weight of all fruits collected for that species I assume? Display with consistent two decimal places.
Table 2: What do the different colors mean? This shows the period when the plant are fruiting and/or when the fruits are attacked by the respective fruit flies? What do the asterisks mean? This table needs some clarification.
172: What is fruit richness?
173: italicize species name
174: see comment above
177-184: italicize species names in this paragraph
177-184: If I am seeing this correctly you only talk about recording pupae and adults in the material and methods section, here you report larval data, where does that come from?
Table 3: The numbers of recovered pupae and sex ratio don’t add up, everything missing did not eclose?
187: remove bold font
Table 4: I would change “host scientific name” into “host plant scientific name” just to make sure it is clear with the fruit flies being the hosts of the parasitoids.
Table 4: What is the column “no. of individuals”? You already have the number of fruit flies and the number of parasitoids (that I would switch to be left of the parasitoid sex ratio).
213: fix the first few words, something is wrong
214-215: Be a bit more specific. You want to say that the majority of the host plants has so far not been recorded from Quitana Roo? Mention numbers.
228-232: This is a long sentence, consider splitting into two.
232: remove comma after “C. edulis”
233: remove comma after “C. microcarpa”
239: This could use a little more. You are saying that plants of that family are an important crop, I would add a sentence pertaining to what it means that these fruits are heavily infested.
242-243: would the expression “reservoir” make sense here?
250: Did you find that fruit fly species in any other host plants in your study?
258: Replace “a” with “the”?
256-258: Any thoughts on why there seem to be fewer of those parasitoid on cultivated plants and how that could be changed?
279: “where it is” instead of “which is”?
Author Response
Please see the attachment,
Thank you

Reviewer 3 Report
Dear Authors,
I acknowledge that due to this formidable diversity, comprehensive inventories of insects are between the most challenging topic for global biodiversity assessment.
You manuscript will be of interest to the readers of the journal.
However, necessary steps in a proper study of insect biodiversity include definition of level of detail and selection of sampling methods.
it could be great if you could explain better the following points of the manuscript:
- Materials and methods/Study site: The survey methodology should be described in more detail. It is not clear how many sampling points were considered for each GPS location or if each location is a point.
- Did you planned a minimum distance between the points?
- Sampling points should be considered in the map (Fig.1)
- For each point, the plant species and samples collected should be mentioned.
- During the monthly sampling activities, did you collect samples at the very same locations or even plants?
- Between the sampling points, did you detected any difference in plant (host) abundance? if yes, should be specified in the ms and related to fruit-fly species and parasitism.
- Results should be adjusted after considering the points above.
- Minor text editing is needed.
To conclude, I recommend that the paper should be accepted for the publication after assessing these points.

Author Response
Please see the attachment,
Thank you

Round 2
Reviewer 2 Report
All my comments were addressed, thank you.
Author Response
Thak you again for your comments, time and attention.
